# *Puccinia triticina* and Salicylic Acid Stimulate Resistance Responses in *Triticum aestivum* Against *Diuraphis noxia* Infestation

**DOI:** 10.3390/plants14030420

**Published:** 2025-01-31

**Authors:** Huzaifa Bilal, Willem Hendrik Petrus Boshoff, Lintle Mohase

**Affiliations:** Department of Plant Sciences, University of the Free State, Bloemfontein 9300, South Africa; boshoffwhp@ufs.ac.za (W.H.P.B.); mohasel@ufs.ac.za (L.M.)

**Keywords:** plant priming, *Puccinia triticina*, salicylic acid, *Diuraphis noxia*, wheat, antioxidants

## Abstract

Wheat plants encounter both biotic and abiotic pressure in their surroundings. Among the biotic stress factors, the Russian wheat aphid (RWA: *Diuraphis noxia* Kurdjumov) decreases grain yield and quality. The current RWA control strategies, including resistance breeding and the application of aphicides, are outpaced and potentially environmentally harmful. Alternatively, priming can stimulate defence responses to RWA infestation. This study investigated the priming potential of two priming agents, avirulent *Puccinia triticina* (*Pt*) isolates and salicylic acid (SA), against RWA infestation. The priming effect of *Pt* isolates and SA in reducing RWA-induced leaf damage and increased antioxidant activities is an indication of defence responses. Selected South African wheat cultivars and Lesotho landraces, grown under greenhouse conditions, were inoculated with *Pt* isolates (UVPt13: avirulent, UVPt26: virulent) and treated with SA at the seedling or booting stages. The leaf damage rating score was used for phenotyping. The antioxidant-mediated defence responses were evaluated in three selected cultivars for further priming investigation. Our results revealed that the priming agents significantly reduced the leaf damage in most cultivars at both growth stages, and UVPt13 and SA priming significantly (*p* ≤ 0.05) increased superoxide dismutase, peroxidase, and ascorbate peroxidase activities. However, catalase activity exhibited a more pronounced decline in plants treated with the UVPt13 isolate. The *Pt* isolate priming was more efficient than the SA application. However, it is crucial to investigate the potential of effectors from the avirulent *Pt* isolate to prime wheat plants for resistance against RWA infestation. This could contribute to developing strategies to enhance crop protection and relieve pest pressure in wheat production.

## 1. Introduction

Russian wheat aphid (RWA) is a devastating pest that causes leaf turgor loss, reduces biomass, inhibit growth, and, in severe infestation, causes plant death [1,2]. It has the potential to reduce grain yield by up to 93% in different wheat producing regions [3]. Current control strategies, which predominantly rely on chemical insecticides, are becoming less effective due to the rapid development of insecticide resistance in RWA populations [4]. Furthermore, these chemicals pose environmental risks, including the pollution of soil and water resources, and have been associated with negative impacts on non-target organisms. An alternative strategy is to develop resistant cultivars through breeding. However, this strategy is relatively slower than the evolution of virulent RWA biotypes [5].

Plant priming is an alternative strategy where plants are exposed to a mild, non-lethal stress or stimulus that enhances their resistance or tolerance to future stresses, such as pests, pathogens, or environmental stressors. This pre-physiological condition enables the plants to react more effectively against biotic and abiotic stressors and improve growth and productivity [6]. Plant priming induced by the application of various chemicals, including plant extracts [7], phytohormones [8], and microorganisms [9] to plants, stimulates various defence responses to stress factors through the increase in antioxidant capacity [10], such as superoxide dismutase (SOD), ascorbate peroxidase (APX), glutathione peroxidase (GPX), monodehydroascorbate reductase, and catalase (CAT). The synthesis of metabolites such as tocopherol and carotenoids [11] also increases the antioxidative capacity and protects from toxic reactive oxygen species (ROS). The glutathione reductase (GR), POD, APX, tocopherol, and CAT further work together with SOD to prevent cell metabolites from H_2_O_2_ [12].

The ROS crosstalk with abscisic acid, jasmonic acid, and the SA signalling pathways as part of the defence mechanism [13]. Despite this, the overproduction of ROS sources cause photooxidative damage to cell metabolites including nucleic acid, proteins, lipids, and carbohydrates [14]. Therefore, plants activate enzymatic and non-enzymatic antioxidants to detoxify overexpressed ROS. Salicylic acid interacts with the transcriptional factors (nonexpressor of the pathogenesis-related protein) to activate SA-arbitrated gene expression [15]. Applying exogenous SA [16] or plant microbes as priming agents activates plant defence responses and stimulates host resistance to future stressors [17]. Ref. [18] revealed that the pre-inoculation of a resistant (SST 347) and susceptible wheat cultivar (SST 356) primed with *Pt* isolate 3SA145 induced a resistance (antixenosis) response to RWASA1 infestation, but they did not evaluate the role of antioxidants as part of the resistance mechanism.

We hypothesise that pre-treatment of wheat plants with avirulent *Pt* isolates or exogenous SA will enhance defence responses against RWA infestation, as evidenced by reduced leaf damage and increased antioxidant enzyme activities. This study aims to evaluate the efficacy of these priming agents in inducing resistance and their potential as part of integrated pest management strategies in wheat production.

## 2. Materials and Methods

The impact of RWA infestation on wheat cultivars primed with *Pt* isolates or treated with SA was evaluated. A selection of South African winter wheat cultivars (PAN 3111, PAN 3118, PAN 3133, PAN 3161, PAN 3368, and SST 356; obtained from the Agricultural Research Council-Small Grain, South Africa; ARC-SG [19]) and Lesotho landraces (Bolane, Makalaote, Mapili, and Tsholoha) were evaluated at the seedling and booting stage. The RWA biotypes were obtained from the ARC-SG institute. Experiments were conducted in the greenhouse under a natural photoperiod and day and night temperatures of 24 ± 1 °C and 18 ± 1 °C, respectively. Experiments were performed in a randomised complete block design (RCBD) fashion.

### 2.1. Phenotyping of Primed and Non-Primed Wheat Plants to RWA Infestation

The plants were raised to the seedling (two-leaf) [20] or booting (Zadoks 45) stages before being inoculated with *Pt* isolates and treated with SA. Wheat plants were treated with a 1.5 mM SA solution, prepared by initially dissolving in 1 mL ethanol, and sprayed using an atomiser spray bottle until there was runoff on leaves. Isolate UVPt13 of leaf rust race 3SA140 is avirulent to *Lr*-3a, -3bg, -3ka, -11 -16, -20, and -30, and virulent to *Lr*-1, -2a, -2b, -2c, -10, -14a, -15, -24, and -26 [21]. UVPt26 race 3SA248 has a wider virulence to most *Lr* genes available, including *Lr*-20 and *Lr*-26 [22], so UVPt13 and UVPt26 were chosen to evaluate the priming effects against RWA infestation. Wheat plants were inoculated with *Pt* urediniospores stored at −80 °C. Before inoculation, the urediniospores were water bath heat shocked at 46 °C for 6 min. The spore mass of 1 mg was suspended in 0.8 mL Soltrol^®^ 130 isoparaffinic oil (Chevron Phillips Chemical Company, The Woodlands, TX, USA). Plants were inoculated using a pressure pump (Vacuubrand^®^ MZ2, Wertheim, Germany) set at 25 KPa with an attached inoculation nozzle in an enclosed inoculation booth pre-rinsed with filtered water [22]. Two days following treatments, plants were infested with RWASA1 or 4 at the rate of 20 adult apterous aphids per plant at the seedling stage or 150 adult apterous aphids per plant at the booting stage. Aphid-induced leaf damage was evaluated using a damage rating scale after ten days of infestation in the seedling stage and fifteen days of infestation at the booting stage. The scale used for the seedling stage ranged from no infestation symptoms to plant death, where 1 = no damage; 2 = small chlorotic spots; 3 = chlorotic spots; 4 = chlorotic splotches; 5 = mild chlorotic streaks; 6 = prominent chlorotic streaks; 7 = severe chlorotic streaking and conduplicate leaf folding; 8 = severe streaks and convolute leaf rolling; 9 = severe streaks and tight leaf rolling; and 10 = plant dead [23]. The scores 1–2 were categorised as highly resistant, 3–4 as resistant, 5–6 as moderately resistant, 7 as moderately susceptible, and 8–10 as susceptible. At the booting stage, leaf damage was assessed using a score ranging from 1 to 4, where 1 = no damage; 2 = chlorotic spots; 3 = longitudinal striping; and 4 = leaf rolling [24]. The scores 1–2 were categorised as resistant, 3 as moderately susceptible, and 4 as susceptible.

Wheat cultivars, including resistant PAN 3161, and PAN 3118 and PAN 3111 which are susceptible to RWASA1 were selected for further evaluation of the effects of *Pt* isolates (UVPt13) and 1.5 mM SA exogenous applications. The treatments were assigned using an RCBD and plant growth conditions were maintained as mentioned previously. Wheat seedlings (Zadok 12: seedling stage) were evaluated to measure antioxidants in primed and non-primed plants. Leaf samples were harvested in liquid nitrogen at different hours post-infestation (hpi: 0, 6, 9, 12, 24, 48, 72, and 96).

### 2.2. Extraction of Superoxide Dismutase, Peroxidase, and Catalase Enzymes

To measure SOD, POD, and CAT activity, total protein was extracted using the method described by [25]. Leaf samples (1 g) were ground and homogenised on ice in a mortar and pestle in a 4 mL extraction buffer. The extraction buffer of 50 mM potassium phosphate (pH 7.0) consisted of 0.004 g of polyvinylpolypyrrolidone (PVPP), 0.1% (*v*/*v*) Triton X-100, 0.04% (*w*/*v*) sodium metabisulfite, and 10 mM ethylenediaminetetraacetic acid (EDTA). The mixture was centrifuged at 17,000 revolutions per minute (rpm) for 15 min at 4 °C using an Allegra X-30, Beckman Coulter (Brea, CA, USA) centrifuge.

### 2.3. Extraction of Ascorbate Peroxidase Enzyme

Ascorbate peroxidase was extracted according to [26]. Ground leaf tissue (0.5 g) was homogenised in pre-cold mortar and pestle in a 5 mL 50 mM potassium phosphate (PP) buffer (pH 7.0) containing 2% (*w/v*) PVPP, 0.1% (*v/v*) Triton X-100 (Sigma-Aldrich, St. Louis, MO, USA), 1 mM ascorbate, and 1 mM EDTA. The leaf tissue paste was centrifuged at 15,000 rpm for 20 min.

### 2.4. Superoxide Dismutase Activity

The reaction mixture of 50 mM PP buffer at pH 7.8, containing 75 µM nitro blue tetrazolium, 13 mM methionine, 2 µM riboflavin, and 0.1 mM EDTA was used to measure the SOD activity. The activity was measured as described by [27]. The reaction mixture was prepared in disposable polystyrene cuvettes containing 970 µL reaction mixture and 30 µL sample extract. The reaction cuvettes (control and sample) were irradiated by placing them under a fluorescent lamp (40 watts, 30 cm below) for 30 min. The reaction’s absorbance (Abs) was measured at 560 nm with a spectrophotometer (Cary-100 UV-VIS, Agilent, Santa Clara, CA, USA). The following formula is used to calculate SOD activity:$$SOD\;specific\;activity\left(\%\;inhibition\;of\;NBT\right)=\frac{Abs\left(Control\right)-Abs\;(Sample)}{Abs\;(Control)}\times \frac{1}{time\left(minutes\right)}\times \frac{1}{\left[Prot\right]}\times dil\;factor\times 100$$$$Dilution\;factor=(\frac{Total\;volume\;in\;cuvette}{Volume\;of\;enzyme\;extract})$$
[Prot]: protein concentration (mg mL^−1^), dil factor: dilution factor.

### 2.5. Peroxidase Activity

The POD activity was measured using the protocol mentioned by [28]. The 1 mL reaction mixture contained 840 µL of 40 mM PP buffer (pH 5.5), 100 µL of 5 mM guaiacol, 0.2 mM EDTA, 10 µL of enzyme extract, and 50 µL of 8.2 mM H_2_O_2_ which initiated the reaction. The change in Abs was recorded at 470 nm using a spectrophotometer for 180 s at 30 °C. The given equation is used to calculate POD activity.$$POD\;specific\;activity=\frac{dil\;factor\times ∆Abs}{\sum \times \left[Prot\right]}\;\mu mol\;tetraguaiacol\;{mg}^{-1}prot\;s^{-1}$$
∑ = tetraguaiacol molar absorptivity (26.6 mM^−1^ cm^−1^), ∆Abs = (change in Abs), dil factor: As indicated in Section 2.4.

### 2.6. Catalase Activity

The CAT activity was assessed by monitoring the breakdown of H_2_O_2_ at 25 °C for 1 min, with a spectrophotometer set to 240 nm [29]. The reaction mixture contained 630 µL of deionised water, 330 µL of 59 mM H_2_O_2_ in 50 mM PP buffer at pH 7.0, and 40 µL of enzyme extract. The activity was calculated as follows:$$CAT\;specific\;activity=\frac{Abs}{\sum }\times \frac{dil\;factor}{\left[Prot\right]}\;µM\;H_{2}O_{2}\;per\;mg\;protein\;per\;minute$$
∑ = H_2_O_2_ molar absorptivity 39.9 M^−1^cm^−1^, dil factor: As indicated in Section 2.4.

### 2.7. Ascorbate Peroxidase Activity

The modified method by [30] was used to determine APX activity. The reaction mixture of 1 mL contained 500 µL of 100 mM PP buffer at pH 7.0, 200 µL of 4 mM H_2_O_2_, 200 µL of 0.68 mM ascorbate, 0.1 mM EDTA and 100 µL of sample extract. A decrease in absorbance resulting from ascorbate oxidation was measured at 290 nm on a spectrophotometer for 1 min. The following equation was used to measure APX activity.$$APX\;specific\;activity=\frac{dil\;factor\times \frac{∆Abs}{∆t}\times \frac{1}{\sum }}{\left[Prot\right]}mM\;cm\;µg\;{mL}^{-1}$$
∆t = reaction time, ∑ = ascorbic acid molar absorptivity (2.8 mM^−1^ cm^−1^), dil factor: As indicated in Section 2.4.

## 3. Results

The analyses of variance (ANOVA) were performed to evaluate the treatment (*Pt* isolates and SA priming) effects on different cultivars at two plant growth stages. The induced leaf damage and specific antioxidative enzyme activities were analysed, and the grouping of treatments was used to evaluate priming efficacy. The ANOVA indicated that treatments, cultivars, and their interactions were highly significant, while replication was nonsignificant (Table 1). The results showed that RWA-induced damage significantly differed in primed and non-primed plants. Biotype 4 (RWASA4) induced severe leaf damage in all the untreated (control) cultivars, except SST 356 and PAN 3161, and was regarded as more damaging than RWASA1. However, *Pt* isolates reduced induced leaf damage by RWASA4, indicating shifts to moderately resistant and resistant responses. Similarly, UVPt13 and UVPt26 pre-inoculation enhanced resistance to RWASA1, shifting the response to a resistant reaction. Salicylic acid application before infestation had a minor impact on wheat responses, reducing RWASA4-induced leaf damage only in two cultivars (PAN 3161 and SST 356); however, it significantly reduced RWASA1-induced leaf damage in most of the wheat cultivars. The reaction categories shifted from susceptible to moderately susceptible (PAN 3118 and Mapili), and moderately susceptible/resistant to resistant (PAN 3133, SST 356, and Bolane).

### 3.1. Phenotyping of Wheat Plants Prepared and Not Prepared for RWA Infestation

#### 3.1.1. Seedling Stage

Phenotyping revealed that RWASA1 infestation caused leaf streaking and leaf rolling at the seedling stage (Figure 1B) in most cultivars. The infestation in SA-primed plants caused leaf streaking and mild chlorosis (Figure 1C). However, SA priming improved the wheat response to RWASA1 from moderately resistant to resistant in cultivars PAN 3133, SST 356, and Bolane (Figure 1A–C; Table 2).

RWASA1 infestation of *Pt* pre-inoculated wheat cultivars also caused mild leaf damage, evident as scattered chlorosis (Figure 1D,E), especially in the South African cultivars. *Pt* inoculation reduced RWASA1-induced leaf damage in PAN 3133 and SST 356, improving the resistance categories from moderately resistant to resistant (Figure 1A,B,D,E; Table 2). Although PAN 3368 and PAN 3161 exhibited resistance with moderate chlorosis in response to RWASA1 infestation, inoculation with *Pt* and SA priming further improved resistance, resulting in only slight chlorosis (Table 2; Figure 1A–E). In contrast, regardless of priming, the Lesotho landraces were severely damaged by RWASA1 infestation (extensive streaking and minor leaf rolling).

All wheat cultivars including Lesotho landraces were susceptible to RWASA4, while SST 356 and PAN 3161 were moderately susceptible (Figure 1F). However, the RWASA4 infestation of *Pt* (UVPt13)-primed PAN 3111, PAN 3118, PAN 3133, PAN 3161, and SST 356 induced mild symptoms, with a damage rating score categorising the cultivars into having a resistant reaction (Figure 1F–I). In parallel, UVPt26 mediated decreases in RWASA4-induced leaf damage as the mean damage scores recorded were lower and were categorised into moderate resistance (PAN 3118 and SST 356) or resistance (PAN 3161, PAN 3133, and PAN 3111). Even SA treatment reduced leaf damage to RWASA4. The cultivars PAN 3133 and PAN 3368 did not show any priming effects. Landrace Mapili showed the highest induced leaf damage, while PAN 3161 was resistant to infestation of both RWA biotypes (Table 2).

#### 3.1.2. Booting Stage

The ANOVA indicated that treatments, cultivars, and their interactions were highly significant. Like the seedling stage, priming and non-priming treatments significantly influenced induced leaf damage at the booting stage. According to the analysis, *Pt* isolates or SA treatment before infestation in the different wheat cultivars lowered the intensity of RWA-induced leaf damage (Table 3).

Russian wheat aphid infestation caused mild chlorosis, striping, and leaf rolling in wheat plants during the booting stage (Figure 2 and Figure 3). Four cultivars (PAN 3111, Makalaote, Bolane, and Tsholoha) expressed longitudinal striping on leaves, showing moderate susceptibility to RWASA1. On the other hand, RWASA4 induced severe streaks and leaf rolling in PAN 3118, indicating susceptibility. Even though Mapili did not show extensive streaking, chlorosis was severe and brown patches indicating dead tissues were evident, showing susceptibility to RWASA4. Salicylic acid application reduced the leaf damage intensity and the damage scores, improving the reaction category to reflect a stronger resistance to RWASA1 (Table 4). PAN 3111 improved from longitudinal striping (moderately susceptible, MS) to chlorotic spots (resistant, R), while Tsholoha, Makalaote, and Mapili did not change their reaction to RWASA1. Salicylic acid also mediated a change in PAN 3118 from leaf rolling (susceptible, S) to longitudinal striping (MS), reducing RWASA4-induced leaf damage. In this instance, Mapili did not change from longitudinal striping but remained moderately susceptible (Table 4).

Similarly, RWA infestation of wheat cultivars pre-inoculated with *Pt* isolates induced less damage on almost all the cultivars. The infestation (RWASA1) of UVPt13 pre-inoculated wheat maintained the resistant reaction in six cultivars. Additionally, aphid-induced damage was less severe in PAN 3111, Tsholoha, and Makalaote, where instead of longitudinal streaks, the leaves displayed chlorotic spots, indicating an improved resistant category (Figure 3). Only Mapili was unresponsive and remained moderately susceptible to RWASA1. In parallel, UVPt13 pre-inoculation reduced RWASA4-induced damage symptoms and scores in almost all cultivars except PAN 3118, where the symptoms improved from leaf rolling to longitudinal striping, indicating moderate susceptibility. All the cultivars primed with UVPt26 maintained their resistance reaction or improved their resistance to RWASA1, except PAN 3111, which remained moderately susceptible. Likewise, all cultivars became resistant to RWASA4, except PAN 3118, which improved to moderate susceptibility.

The landraces, except Mapili, and the cultivars, except PAN 3118, were resistant to RWASA4 during the booting stage. PAN 3118 remained moderately susceptible to RWASA4 infestation despite the prior treatments, while the response of Mapili improved to resistant after pre-inoculation with the *Pt* isolates (Figure 3).

Pre-inoculation with *Pt* isolates reduced the intensity of leaf damage more than SA application at both growth stages. On the flag leaves, the *Pt* isolates reduced infestation damage in most cultivars tested. The two *Pt* isolates could not significantly reduce RWASA4-induced leaf damage in PAN 3118.

### 3.2. Growth Stage Correlation Analysis of Treatments

The growth stage correlation analyses of the priming and infestation treatments of wheat were moderately positive (r = 0.57). The priming and control infestation treatments showed generally strong positive and significant correlations, with the exceptions of 1.5 mM SA + RWASA4 (0.43), UVPt13 + RWASA4 (0.06), and UVPt26 + RWASA4 (0.125). Wheat plants primed by UVPt13 and infested with RWASA1 resulted in the highest positive and significant correlation (0.74).

Hence, correlation analysis revealed that wheat cultivars showed correlated responses at both growth stages. Further evaluation of antioxidant capacity was performed at the seedling stage using UVPt13 as a priming agent and RWASA1 biotype infestation for further evaluation.

### 3.3. Superoxide Dismutase

The ANOVA for SOD showed that primed infested and control plants expressed significantly different activity (Table 5). The SOD activity at different hpi (time) was also significantly different. The replications were nonsignificant. The grouping of means showed that priming agents influenced SOD activity. Priming by UVPt13 induced the highest SOD activity in the wheat cultivars under study, than priming by SA and positive control (RWASA1 infestation). PAN 3111 had a higher SOD activity than the other cultivars, and the grouping of treatments mean values did not show different groups (Table 6). Wheat cultivar PAN 3161 (resistant) showed a gradual increase in SOD activity. However, PAN 3111 showed variant responses over time. A steady peak of SOD activity at 12 and 24 hpi was noticed in PAN 3118 and PAN 3111 (Figure 4).

### 3.4. Peroxidase

The ANOVA showed that priming treatments significantly influenced the POD activity in the wheat cultivars, except PAN 3111 (Table 7). UVPt13 priming induced the highest POD activity of all other treatments. The effect of the duration of infestation (time) was significantly different as well. However, biological repeats did not show significantly different activity. The grouping of means revealed that infestation of wheat cultivars primed with either UVPt13 or SA induced a higher POD activity than in plants infested without priming treatments (Table 8). Among the priming agents, UVPt13 showed more efficiency than SA priming, as POD activity was higher. Plants primed with UVPt13 and infested with RWASA1 exhibited higher POD activity than the controls (without priming). Primed (UVPt13) PAN 3161 showed pronounced activity in the later hours of infestation (72 and 96 hpi; Figure 5).

### 3.5. Catalase

Like SOD, POD, and APX, CAT also showed significantly different activities in wheat plants under different treatments. Biological repeats also showed significantly different CAT activity in PAN 3111 (Table 9). The wheat cultivar PAN 3118 had little variation throughout the sampling times. Strangely, UVPt13 reduced CAT activity to even lower than the positive control without infestation (Table 10). However, SA priming induced higher CAT activity in PAN 3111 during early hours of infestation (Table 10, Figure 6).

Throughout the trial, the RWASA1 infestation of UVPt13-primed plants suppressed CAT activity in all the cultivars. UVPt13 priming induced the steepest decline in CAT activity, while an infestation of SA-primed wheat induced a significant increase at 96 h after infestation in PAN 3161 and 3118, somewhat delaying effective resistance to infestation (Figure 6).

### 3.6. Ascorbate Peroxidase

Like SOD and POD, APX was significantly different in wheat cultivars under different treatments. Biological repeat showed consistency of APX activity (Table 11). Wheat plants without priming and infestation showed the lowest APX activity, while plants primed by UVPt13 showed the highest activity (Table 12). Infestation of SA-primed plants showed the highest APX activity, followed by UVPt13 priming in PAN 3161 and PAN 3111. UVPt13-primed plants increased in APX activity to reach a peak at 6 hpi, which dropped at 9 hpi and was maintained in the later hours of sampling (Figure 7). PAN 3118 showed similar APX activity in infested and primed (UVPt13) infested plants in the later hours. Infestation of UVPt13-primed plants induced higher activity in the wheat cultivars under study than those primed by SA (Table 12, Figure 7).

## 4. Discussion

Eight out of ten wheat cultivars were susceptible to RWASA4, showing severe induced leaf damage at the seedling stage, while the remaining were susceptible to RWASA1. Unexpectedly, at the booting stage, more cultivars were resistant (six and eight) to RWASA1 and -4, respectively. Even though stage-related resistance to RWA has not been reported for the cultivars used in this study, we cannot exclude adult-stage resistance, as it occurs in some wheat–pathogen interactions [31]. The main differences in responses to infestation at both growth stages were evident in the primed treatments. Cultivars primed with *Pt* isolates or with SA improved their defence responses to RWA infestation, evidenced by reduced leaf damage symptoms (Table 2 and Table 4). The correlation analysis showed positive correlations of the priming effect at both plant growth stages. This suggests that plants can be primed at either the seedling or booting stage against RWA infestation.

Pre-inoculation with *Pt* isolates changed wheat cultivars’ reaction to the two aphid biotypes at the different growth stages. In another study, inoculation with four different endophytic fungi increased resistance in wheat plants against the take-all disease (*Gaeumannomyces graminis* var. *tritici*) [32]. The current findings support the hypothesis that priming with avirulent *Pt* isolates can enhance host resistance to RWA infestation. Likewise, [18] revealed that *Pt* isolates can activate antibiotic responses in wheat plants to protect against RWA infestation. The literature also reports that plant growth-promoting fungi (*Penicillium chrysogenum, Aspergillus falvus, A. niger, P. citrinum, and T. koningiopsis*) induce systemic resistance [33], evidenced by minor physical damage in wilt disease (*Rhizoctonia solani*; [34]. Priming by avirulent *Pt* isolates (UVPt13) suppressed RWA-induced damage in more wheat cultivars than SA, indicating more substantial priming effects by avirulent *Pt* isolates.

Plant hormones like SA mediate defence responses [35] and decrease aphid (*Lipaphis erysimi*) infestations and populations on mustard plants [36]. Furthermore, another study has revealed that SA application induced resistance in wheat plants and ultimately reduced the aphid population size [37]. Jasmonic acid [38] and SA [16] activate defence responses in wheat plants, which seemingly deterred *Sitobian avenae* colonisation and inhibited feeding. Exogenous SA application in rice disrupted *Oebalus pugnax* nymphal development and reduced the rice stink bug population [39]. In this study, the foliar application of SA reduced damage inflicted by RWA infestation at both growth stages, apparently through mediating the expression of host defence responses. The susceptible cultivars became moderately resistant or resistant.

Previously, *Fusarium*, *Trichoderma* [33], and *Piriformospora* fungi have been introduced as endo-fungi to activate induced systemic resistance in crops against pests and pathogens [40]. *Trichoderma* is commercially available as a defence activator with different brand names [41]. Russian wheat aphid infestation in wheat production has been managed by host plant resistance, where specific *Diuraphis noxia* resistance genes were incorporated as sources of resistance. This strategy succeeded until resistance-breaking biotypes were discovered [42]. There are currently limited wheat sources in South Africa with resistance to the most damaging RWASA5, except the *Dn*7 gene [42]. This study contributes to the search for alternative strategies to manage RWA infestation in wheat. This study revealed that wheat cultivars can be primed using avirulent *Pt* isolates or SA to stimulate defence responses and reduce the induced RWA feeding effects. The results, nonetheless, require further studies to establish the biocontrol effects on wheat yield under field conditions.

The priming agents SA and avirulent *Pt* isolates enhanced antioxidant activities in wheat plants infested by RWA. Furthermore, *Pt* isolates showed improved priming effects by enhancing stronger antioxidant activities than exogenous SA. UVPt13-primed wheat cultivars induced higher activities of SOD, POD, and APX and reduced CAT activity. Induced antioxidant activities indicated an enhanced defence response in wheat plants against RWA infestation.

SA is a plant hormone and phenolic compound and contributes as a signalling molecule to initiate defence responses [13], such as systemic acquired resistance (SAR). It also contributes as an antioxidant to detoxify ROS [43] and activates other antioxidants as well. Likewise, SA application induces SAR in plants and reduces the devastating effects of biotic stressors [44,45]. The foliar application of SA induced the high antioxidant activities in wheat plants infested by RWASA1. Likewise, [45] reported that SA application induced SOD and inhibited CAT activity in wheat plants infected by *F. graminearum*. Activated antioxidant activities are an indication of ROS regulation; ROS are important signalling molecules that activate some defence responses [46]. Our results agree with [47], who revealed that wheat seed priming by SA enhances resistance to *F. graminearum* infection in plants. Furthermore, they examined an increase in polyphenol oxidase, POD, and SOD, and the accumulation of a high level of phenylalanine ammonia-lyase (PAL) mRNA, chitinase, and β-1,3-glucanase. Their findings align with certain results of this study, indicating that the POD and APX activities in PAN 3111 and PAN 3161 (Table 8 and Table 12) were elevated in plants treated with SA. The scavenging of ROS helps protect plants from oxidative stress caused by pests [25].

*Puccinia triticina,* on the other hand, severely reduces wheat plant growth and yield [48], though avirulent pathotypes such as the UVPt13 isolate have minor effects on wheat. Wheat plants primed with avirulent *Pt* isolates (UVPt13) boost their resistance to RWA infestation by significantly increasing their enzymatic antioxidant levels compared to infested controls. Likewise, [49] reported that *Pt* inoculation of resistant wheat cultivars induced antioxidant activities, and increased the concentrations of flavonoid and phenolic compounds. In [50], the authors conducted a comparison of resistant and susceptible wheat genotypes inoculated with *Pt* isolates, revealing that the resistant genotypes exhibited significantly enhanced enzymatic antioxidative activities after inoculation, including SOD, CAT, and PAL, when contrasted with their non-inoculated cultivars. The augmented activity of SOD serves as a resistance marker to biotic stresses. Likewise, resistant wheat plants infected with *Puccinia striiformis* demonstrated a notable increase in the levels of enzymatic antioxidants (GR, GPX, and APX) in comparison to the susceptible plants [51]. Thus, the incompatible interaction between *Pt* isolates and wheat plants elevates antioxidant activities and strengthens defence responses. However, the incompatible interaction of avirulent *Pt* isolates and wheat plants could enhance defence responses and be exploited to counter RWA infestation. Comparable results have been stated by [18], who found that avirulent *Pt* isolates (3SA145) enhanced antixenosis and proteomic expression in wheat cultivars to RWASA1 infestation. The authors of [52] also reported that plant growth-promoting bacteria and arbuscular mycorrhizal fungi increase phenotypic and proteomic responses upon leaf pathogen infection.

At high concentrations of H_2_O_2_, CAT functions catalytically, with H_2_O_2_ acting as a donor and acceptor. The scavenging of H_2_O_2_ by CAT enzymes at intracellular or intracellular organelles is specific to genotypes and stress, influencing the processing of defence signals in plants. Antioxidants such as CAT play dire roles in plant defence against different stress factors, including aphid infestation [53]. Even though CAT contributes to the resistance mechanism, our results showed a significant reduction in plants primed by *Pt* isolates [54]. In contrast to our results, CAT activity increased in *Vigna mungo* plants infested by white fly (*Bemisia tabaci*) [55]. A possible reason for the reduced CAT activity during the *Pyricularia oryzae* infection of wheat could be the high induction of H_2_O_2_ [56]. The decrease in CAT activity may be attributed to increased proteolysis induced by oxidative stress [57]. Additionalsly, it was reported that many peroxisomal proteins, including CAT, glucose-6-phosphate dehydrogenase, and glycolate oxidase, underwent endoproteolytic degradation. In this study, the high production of H_2_O_2_ may lead to the proteolysis of CAT activity in plants primed by *Pt* isolates, with evidence of high production with increased SOD, POD, and APX activities.

The findings, grounded on enzymatic antioxidative (SOD, POD, and APX) activities, demonstrated that avirulent *Pt* isolate inoculation and the exogenous application of SA could effectively prime wheat plants, boosting host resistance and mitigating the severity of RWASA1-induced effects. This study highlighted that the avirulent *Pt* isolate (UVPt13) was a superior priming agent compared to SA. Although only one RWA biotype was examined thoroughly, priming wheat plants with avirulent *Pt* isolates and SA presents a promising alternative strategy for managing RWA. However, supplementary research is essential to explore the broader applicability of priming across various RWA biotypes, intending to stimulate both horizontal and vertical resistance.

## 5. Conclusions

*Puccinia triticina* and salicylic acid priming significantly enhanced the defence response in wheat plants infested with RWA infestation. Comparatively, the priming agents showed that *Pt* isolates were more effective than SA in conferring resistance to the aphid biotypes. Notably, the *Pt* isolate demonstrated superior antioxidant potential compared to SA priming. These findings underscore the pivotal role of enzymatic antioxidants in mediating tolerance to RWA infestation. Given the varied responses among wheat cultivars, future investigations should delve into the specificity of priming mechanisms and their impact on antioxidant expression in wheat. Nevertheless, this study reveals crucial insights into the priming-mediated mechanisms against RWA infestation. Furthermore, there is a pressing need to assess the efficacy of avirulent *Pt* isolate priming against different RWA biotypes under field conditions or priming with effectors from avirulent *Pt* isolates to ascertain its practical applicability.

## Figures and Tables

**Figure 1 plants-14-00420-f001:**
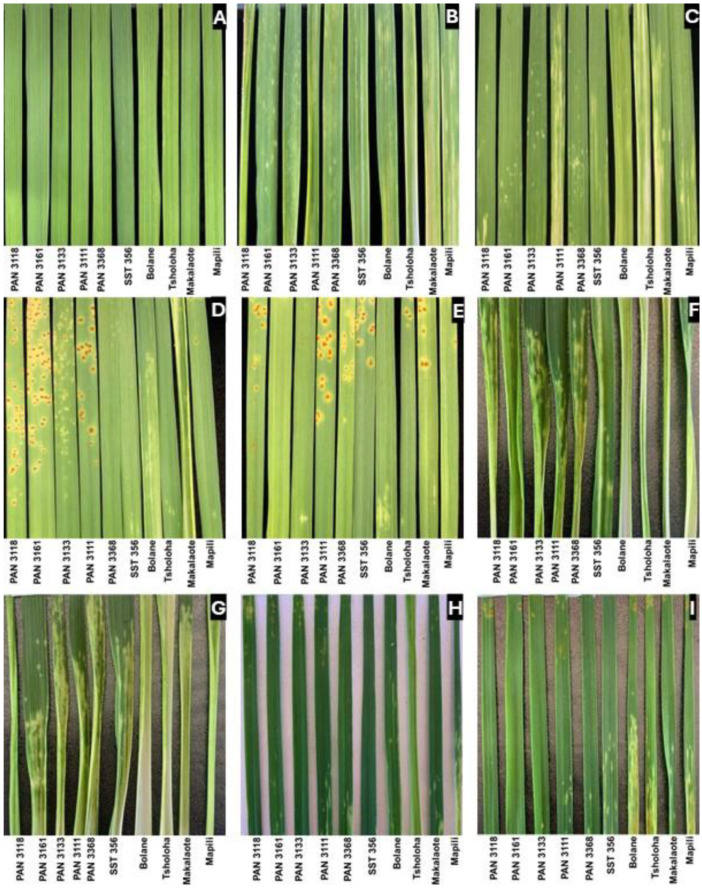
Russian wheat aphid infestation on primed [*Puccinia triticina* isolates and salicylic acid] and non-primed cultivars at the seedling stage. The images represent the second leaf 10 days after infestation. (**A**) Control (without priming and infestation); (**B**) RWASA1 (without priming); (**C**) 1.5 mM SA + RWASA1; (**D**) UVPt13 + RWASA1; (**E**) UVPt26 + RWASA1; (**F**) RWASA4 (without priming); (**G**) 1.5 mM SA + RWASA4; (**H**) UVPt13 + RWASA4; and (**I**) UVPt26 + RWASA4.

**Figure 2 plants-14-00420-f002:**
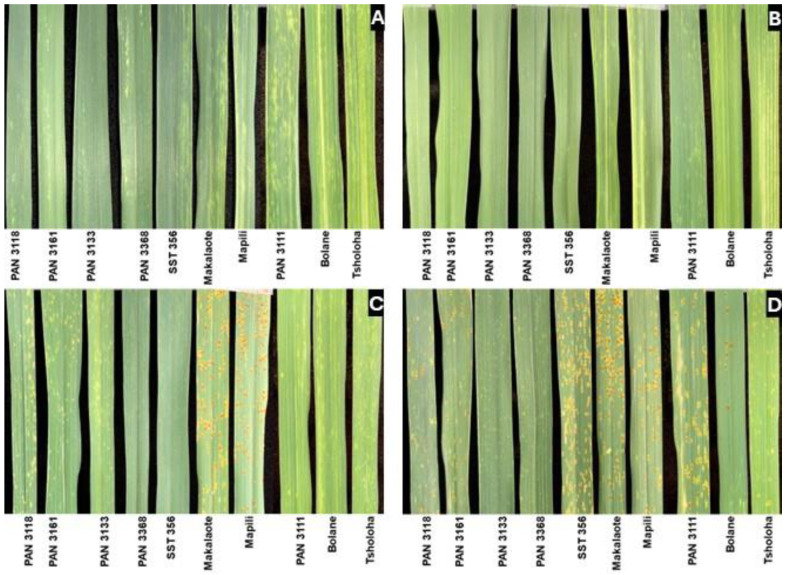
Russian wheat aphid biotype 1 infestation on primed [*Puccinia triticina* isolates and salicylic acid] and non-primed wheat cultivars at the booting stage. The images represent flag leaves 15 days after infestation. (**A**) RWASA1; (**B**) 1.5 mM SA + RWASA1; (**C**) UVPt13 + RWASA1; and (**D**) UVPt26 + RWASA1.

**Figure 3 plants-14-00420-f003:**
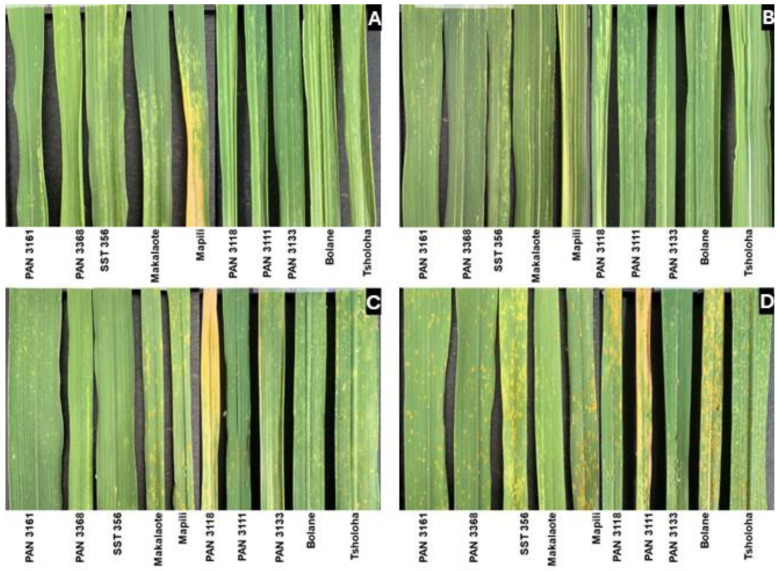
Russian wheat aphid biotype 4 infestation on primed [*Puccinia triticina* isolates and salicylic acid] and non-primed wheat cultivars at the booting stage. The images represent flag leaves 15 days after infestation. (**A**) RWASA4; (**B**) 1.5 mM SA + RWASA4; (**C**) UVPt13 + RWASA4; and (**D**) UVPt26 + RWASA4.

**Figure 4 plants-14-00420-f004:**
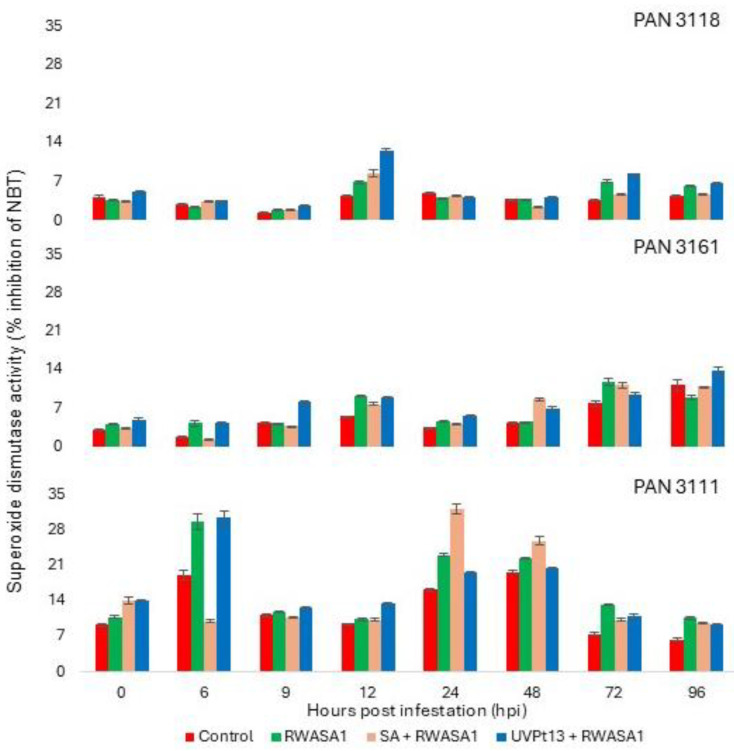
Russian wheat aphid biotype 1 infestation effects on superoxide dismutase activity in primed [UVPt13 and SA] and non-primed plants at the seedling stage. The graph represents means ± standard errors.

**Figure 5 plants-14-00420-f005:**
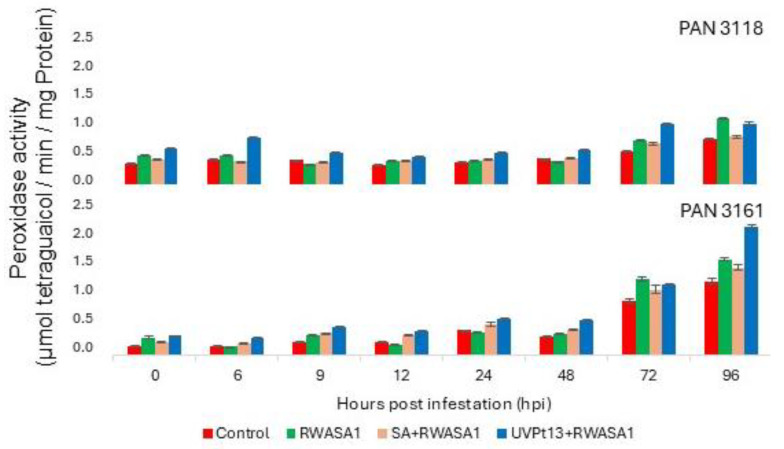
Russian wheat aphid biotype 1 infestation effects on peroxidase activity in primed [UVPt13 and SA] and non-primed plants at the seedling stage. The graph represents means ± standard errors.

**Figure 6 plants-14-00420-f006:**
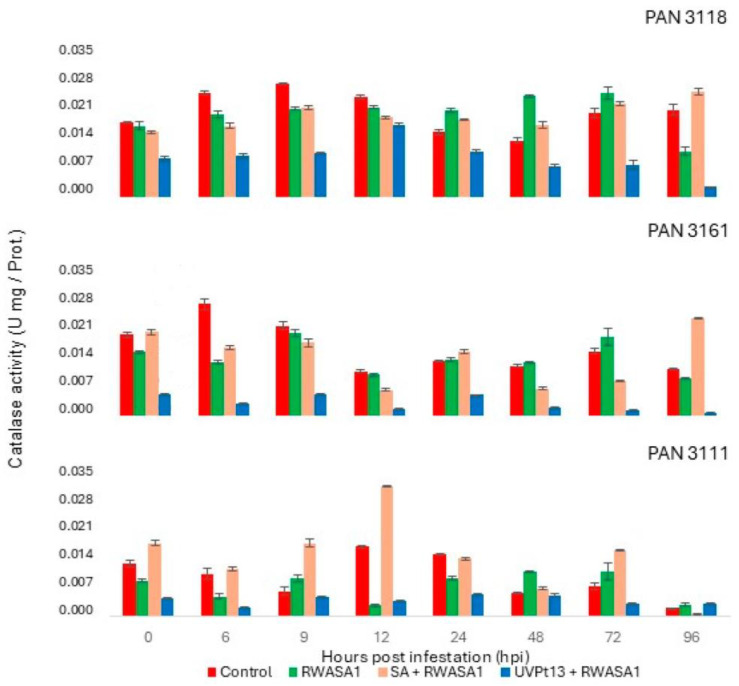
Russian wheat aphid biotype 1 infestation effects on catalase activity in primed [UVPt13 and SA] and non-primed plants at the seedling stage. The graph represents means ± standard errors.

**Figure 7 plants-14-00420-f007:**
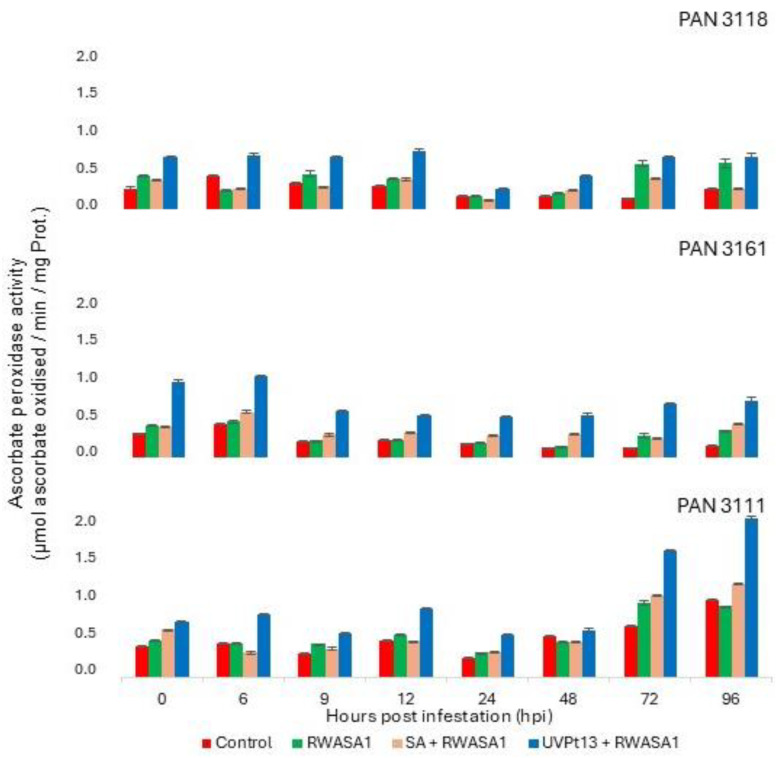
Russian wheat aphid biotype 1 infestation effects on ascorbate peroxidase activity in primed [UVPt13 and SA] and non-primed plants at the seedling stage. The graph represents means ± standard errors.

**Table 1 plants-14-00420-t001:** Analysis of variance of primed [*Puccinia triticina* race isolates (UVPt13 and UVPt26) and salicylic acid] wheat cultivars infested by the South African Russian wheat aphid biotypes (RWASA1 and RWASA4) at the seedling stage.

SOV	Df	*p*-Value	SOV	Df	*p*-Value
Treatment	7	<0.0001	Replication	1	0.3171
Cultivar	9	<0.0001	Treatment × Cultivar	63	<0.0001

*p* ≤ 0.01 = highly significant, *p* ≤ 0.05 = significant, SOV = source of variation, Df = degree of freedom.

**Table 2 plants-14-00420-t002:** Homogeneous grouping of wheat cultivars and treatments at the seedling stage. Values represent induced leaf damage scores.

Homogeneous Grouping of Cultivars					
Cultivars	Mean Values	Groups	Cultivars	Mean Values	Groups
Mapili	7.85	A	PAN 3368	5.13	EF
Makalaote	7.69		PAN 3133	5.01	FG
Tsholoha	7.33	B	PAN 3111	4.76	GH
Bolane	6.80	C	SST 356	4.05	I
PAN 3118	5.56	D	PAN 3161	3.33	J
Homogeneous Grouping of Treatments					
Treatments	Mean values	Groups	Treatments	Mean values	Groups
RWAS4	8.76	A	UVPt13 + RWASA4	5.84	D
SA + RWASA4	7.97	B	SA + RWASA1	4.93	E
UVPt26 + RWASA4	6.37	C	UVPt13 + RWASA1	3.35	F
RWASA1	6.06	D	UVPt26 + RWASA1	2.74	G

*n* for cultivars = 16, *n* for treatments = 20. Mean values: 1–4 = resistant; 5–6 = moderately resistant; 7 = moderately susceptible; and 8–10 = susceptible.

**Table 3 plants-14-00420-t003:** Analysis of variance in primed [*Puccinia triticina* isolates (UVPt13 and UVPt26) and salicylic acid (1.5 mM)] wheat cultivars infested by the South African Russian wheat aphid biotypes RWASA1 and RWASA4 at the booting stage.

SOV	Df	*p*-Value	SOV	Df	*p*-Value
Treatment	7	<0.0001	Replication	1	0.3568
Cultivar	9	<0.0001	Treatment × Cultivar	63	<0.0001

*p* ≤ 0.01 = highly significant, *p* ≤ 0.05 = significant, SOV = source of variation, Df = degree of freedom.

**Table 4 plants-14-00420-t004:** Homogeneous grouping of wheat cultivars and treatments at the booting stage. Values represent induced leaf damage scores.

Homogeneous Grouping of Cultivars					
Cultivars	Mean Values	Groups	Cultivars	Mean Values	Groups
Mapili	2.56	A	Bolane	2.10	C
Tshaloha	2.31	B	SST 356	1.53	D
PAN 3118	2.16	C	PAN 3133	1.35	E
Makalaoti	2.13		PAN 3111	1.33	
PAN 3368	2.13		PAN 3161	1.00	F
Homogeneous Grouping of Treatments					
Treatments	Mean values	Groups	Treatments	Mean values	Groups
RWASA4	2.21	A	SA + RWASA1	1.80	D
SA + RWASA4	2.08	B	UVPt13 + RWASA4	1.80	
UVPt26 + RWASA4	1.99	C	UVPt26 + RWASA4	1.54	E
RWASA1	1.97		UVPt13 + RWASA1	1.52	

*n* for cultivars = 16, *n* for treatments = 20. Mean values: 1–2 = resistant; 3 = moderately susceptible; and 4 = susceptible.

**Table 5 plants-14-00420-t005:** Analysis of variance of superoxide dismutase activity in primed [*Puccinia triticina* (UVPt13) and salicylic acid (1.5 mM)] and non-primed wheat cultivars infested by the South African Russian wheat aphid biotype 1 (RWASA1).

SOV	Df	*p*-Values		
		PAN 3118	PAN 3161	PAN 3111
Treatment	3	0.0035	0.058	0.0213
Replication	2	0.400	0.205	0.4890
Time (hpi)	7	<0.0001	<0.0001	<0.0001
Treatment × Time	21	0.2401	0.9432	0.0072

*p* ≤ 0.01 = highly significant, *p* ≤ 0.05 = significant, SOV = source of variation, Df = degree of freedom.

**Table 6 plants-14-00420-t006:** Homogeneous grouping of the South African Russian wheat aphid (RWASA1)-induced superoxide dismutase activity in salicylic acid-treated and *Puccinia triticina*-inoculated wheat seedlings.

PAN 3118			PAN 3161			PAN 3111		
Treatments	Means	Group	Treatments	Means	Group	Treatments	Means	Group
UVPt13 + RWASA1	5.91	A	UVPt13 + RWASA1	7.78	A	UVPt13 + RWASA1	16.50	A
RWASA1	4.51	B	RWASA1	6.42	BC	RWASA1	16.34	
SA + RWASA1	4.19		SA + RWASA1	6.34		SA + RWASA1	15.35	
Control	3.73		Control	5.16	D	Control	12.31	B

*n* = 24, means grouping at *p* ≤ 0.05.

**Table 7 plants-14-00420-t007:** Analysis of variance of the South African Russian wheat aphid (RWASA1)-induced peroxidase activity in primed [*Puccinia triticina* (UVPt13) and salicylic acid (1.5 mM)] wheat cultivars.

SOV	Df	*p*-Values		
		PAN 3118	PAN 3161	PAN 3111
Treatment	3	<0.0001	<0.0001	0.8041
Replication	2	0.5821	0.0062	0.2106
Time (hpi)	7	<0.0001	<0.0001	<0.0001
Treatment × Time	21	0.1970	0.2473	0.9731

*p* ≤ 0.01 = highly significant, *p* ≤ 0.05 = significant, SoV = source of variation, Df = degree of freedom.

**Table 8 plants-14-00420-t008:** Homogeneous grouping of South African Russian wheat aphid (RWASA1)-induced peroxidase activity in salicylic acid and *Puccinia triticina* pre-inoculated wheat cultivars.

PAN 3118			PAN 3161			PAN 3111		
Treatments	Means	Group	Treatments	Means	Group	Treatments	Means	Group
UVPt13 + RWASA1	0.689	A	UVPt13 + RWASA1	0.829	A	UVPt13 + RWASA1	2.10	A
RWASA1	0.540	BC	SA + RWASA1	0.642	B	SA + RWASA1	1.90	
SA + RWASA1	0.488	BCD	RWASA1	0.630		RWASA1	1.72	
Control	0.449	CD	Control	0.498	C	Control	1.67	

*n* = 24, means grouping at *p* ≤ 0.05.

**Table 9 plants-14-00420-t009:** Analysis of variance of catalase activity induced by South African Russian wheat aphid biotype 1 (RWASA1) infestation in primed [*Puccinia triticina* race isolate (UVPt13) and salicylic acid (1.5 mM)] and non-primed wheat cultivars.

SOV	Df	*p*-Values		
		PAN 3118	PAN 3161	PAN 3111
Treatment	3	<0.0001	<0.0001	<0.0001
Replication	2	0.4220	0.0199	0.6143
Time (hpi)	7	0.1786	0.0011	0.0022
Treatment × Time	21	0.2306	0.0550	0.0289

*p* ≤ 0.01 = highly significant, *p* ≤ 0.05 = significant, SOV = source of variation, Df = degree of freedom.

**Table 10 plants-14-00420-t010:** Homogeneous grouping of Russian wheat aphid (RWASA1)-induced catalase activity of wheat cultivars pre-treated with salicylic acid or inoculated with *Puccinia triticina* isolate UVPt13.

PAN 3118			PAN 3161			PAN 3111		
Treatments	Means	Group	Treatments	Means	Group	Treatments	Means	Group
Control	0.0195	A	Control	0.017	A	SA + RWASA1	0.014	A
RWASA1	0.0190		SA + RWASA1	0.015		Control	0.009	BC
SA + RWASA1	0.0186		RWASA1	0.014		RWASA1	0.007	CD
UVPt13 + RWASA1	0.008	B	UVPt13 + RWASA1	0.003	B	UVPt13 + RWASA1	0.004	D

*n* = 24, means grouping at *p* ≤ 0.05.

**Table 11 plants-14-00420-t011:** Analysis of variance of South African Russian wheat aphid biotype (RWASA1)-induced ascorbate peroxidase activity in primed [*Puccinia triticina* race isolate (UVPt13) and salicylic acid (1.5 mM)] wheat cultivars.

SOV	Df	*p*-Values		
		PAN 3118	PAN 3161	PAN 3111
Treatment	3	<0.0001	<0.0001	<0.0001
Replication	2	0.0619	0.4272	0.4092
Time (hpi)	7	0.0021	<0.0001	<0.0001
Treatment × Time	21	0.6928	0.6839	0.6712

*p* ≤ 0.01 = highly significant, *p* ≤ 0.05 = significant, SOV = source of variation, Df = degree of freedom.

**Table 12 plants-14-00420-t012:** Homogeneous grouping of South African Russian wheat aphid (RWASA1)-induced ascorbate peroxidase activity in salicylic acid-treated and *Puccinia triticina*-inoculated wheat cultivars.

PAN 3118			PAN 3161			PAN 3111		
Treatments	Means	Group	Treatments	Means	Group	Treatments	Means	Group
UVPt13 + RWASA1	0.051	A	UVPt13 + RWASA1	0.06	A	UVPt13 + RWASA1	0.1156	A
RWASA1	0.033	BC	SA + RWASA1	0.030	BC	SA + RWASA1	0.0672	B
SA + RWASA1	0.025	CD	RWASA1	0.024	CD	RWASA1	0.0667	
Control	0.022	D	Control	0.018	DE	Control	0.0567	

*n* = 24, means grouping at *p* ≤ 0.05.

## Data Availability

Data are contained within the article.

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
