# Peer review of "Puccinia triticina and Salicylic Acid Stimulate Resistance Responses in Triticum aestivum Against Diuraphis noxia Infestation"

_plants, 2025, doi:10.3390/plants14030420_

Round 1
Reviewer 1 Report
Comments and Suggestions for Authors
This study investigated the priming potential and efficacious fallouts of two
priming agents, avirulent Puccinia triticina (Pt) isolates and salicylic acid (SA)
against RWA infestation. The priming effect of Pt isolates and SA in reducing
RWA-induced leaf damage and increased antioxidant activities as an indication
of defence responses. Selected South African wheat cultivars and Lesotho
landraces, grown under greenhouse conditions, were inoculated with Pt isolates
(UVPt13: avirulent, UVPt26: virulent) and treated with SA at the seedling or booting
stages. The leaf damage rating score was used for phenotyping. The
antioxidant-mediated defence responses were evaluated in three selected cultivars
for further priming investigation. Our results revealed that priming agents
significantly reduced the leaf damage in most cultivars at both growth stages,
and UVPt13 and SA priming significantly (P≤0.05) increased superoxide dismutase,
peroxidase, and ascorbate peroxidase activities. However, catalase activity
exhibited a more pronounced decline in plants treated with UVPt13 isolates.
The Pt isolate priming was more efficient than the SA application. The ability of
the Pt isolates and SA to reduce RWA infestation could contribute to developing
strategies to enhance crop protection and relieve pest pressure in wheat
production.
Some limitations should be improved as following:
1. in the section 2.1, Leaf rust inoculation protocols previously described for
seedlings and plants at the booting stage [20] were used. The description is too
simply. The authors should describe the spore concentration, SA concentration and
inoculation amount in deatails.
2. Table 2-4 and figuture 4-7, significant difference of all the data should be detected 。
3. Line 492, "they could be used as part of alternative strategies for crop 492
protection in plants instead of direct chemical pesticides" cannot be accepted. Because
Puccinia triticina significantly caused disease symptom in wheat. The authors should
correct the conclusion and the abstract.
Author Response
1. in the section 2.1, Leaf rust inoculation protocols previously described for
seedlings and plants at the booting stage [20] were used. The description is too
simply. The authors should describe the spore concentration, SA concentration and
inoculation amount in deatails.
Response: mentioned and highlighted with red font
2. Table 2-4 and figuture 4-7, significant difference of all the data should be detected
Response: changes table 2-4 with LSD grouping tables
3. Line 492, "they could be used as part of alternative strategies for crop 492
protection in plants instead of direct chemical pesticides" cannot be accepted. Because
Puccinia triticina significantly caused disease symptom in wheat. The authors should
correct the conclusion and the abstract.
Response: Fixed it
Reviewer 2 Report
Comments and Suggestions for Authors
Manuscript Review General Comments: The manuscript requires significant revision to align with standard guidelines for structure, reference citations, and clarity. Below are detailed comments and recommendations for improvement: Reference Citations: Ensure all references follow the journal’s citation guidelines consistently. This includes formatting in the text and reference list. Section Alignment: The subtitles in the methodology, results, and discussion sections are not aligned, which disrupts the logical flow of the manuscript. The lack of corresponding sections makes it difficult to clearly follow the hypothesis, objectives, methodology, results, and conclusions. This structure should be revised to ensure coherence. Specific Comments: Line 46: Provide a full explanation of ROS (Reactive Oxygen Species) before its first mention. Clearly describe its role in biological systems, distinguishing between its signaling functions and potential oxidative damage. Line 118: The methodology for measuring SOD ( superoxide dismutase) activity lacks sufficient citation and explanation. Include a brief but clear description of the protocol used, referencing established methodologies. Line 127: Clarify why the protein factor is included in the formula for calculating SOD activity. Justify its necessity and relevance in the analysis. Line 139: Provide a detailed description of the enzyme extract preparation. Include the source, buffer composition, and extraction procedure. Line 144: Similar to SOD, briefly describe the methodology used to measure APX (Ascorbate Peroxidase) activity. Include specific assay conditions and citations to improve reproducibility. Line 154: Correct the term protein, ensuring that terminology matches the context and methodology described. Line 296: Define hpi (hours post-infection) the first time it appears. This will ensure readers unfamiliar with the abbreviation understand its meaning immediately. Additional Notes: Ensure uniform formatting of subsections and adhere to the journal’s structure guidelines for the methodology, results, and discussion sections. Consider rephrasing ambiguous sentences and simplifying overly complex phrasing to enhance readability. Verify all numerical data, figures, and tables for consistency and accuracy. Conclusion: The manuscript has potential but requires substantial revision to improve clarity, alignment, and adherence to guidelines. Addressing these issues will significantly enhance the readability and scientific rigor of the work.

Author Response
Line 46: Provide a full explanation of ROS (Reactive Oxygen Species) before its first mention. Clearly describe its role in biological systems, distinguishing between its signaling functions and potential oxidative damage.
Response: I did. thanks
Line 118: The methodology for measuring SOD ( superoxide dismutase) activity lacks sufficient citation and explanation. Include a brief but clear description of the protocol used, referencing established methodologies
Response: Highlighted and improved
Line 127: Clarify why the protein factor is included in the formula for calculating SOD activity. Justify its necessity and relevance in the analysis
Response: I need to divide by protein concentration to measure specific activity.
Line 139: Provide a detailed description of the enzyme extract preparation. Include the source, buffer composition, and extraction procedure. Line 144: Similar to SOD, briefly describe the methodology used to measure APX (Ascorbate Peroxidase) activity. Include specific assay conditions and citations to improve reproducibility.
Response: I have improved and mentioned appropriate citation
Line 154: Correct the term protein, ensuring that terminology matches the context and methodology described.
Response: Corrected. Thanks
Define hpi (hours post-infection) the first time it appears. This will ensure readers unfamiliar with the abbreviation understand its meaning immediately.
Response: Did. Thanks
I have improved the overall manuscript. Thanks

Reviewer 3 Report
Comments and Suggestions for Authors
The study explores the use of Puccinia triticina isolates and salicylic acid as priming agents to enhance wheat resistance against Russian wheat aphid infestation, which is an innovative approach in the context of pest management. The research involves a comprehensive analysis of defense responses in multiple wheat cultivars, indicating a substantial workload. The methodology is generally sound, with a clear description of the experimental design and treatments. Figures are informative but could be improved for clarity and consistency. The following are suggested revisions.
1. The introduction does not provide sufficient background on the global impact of Russian wheat aphid (RWA) infestations on wheat production. It would be beneficial to include statistics or references to highlight the economic and agricultural significance of this issue.
2. Do you mean? Current control strategies, which predominantly rely on chemical insecticides, are becoming less effective due to the rapid development of insecticide resistance in RWA populations. Furthermore, these chemicals pose environmental risks, including pollution of soil and water resources, and have been associated with negative impacts on non-target organisms.
3. The introduction to plant priming is somewhat abrupt and lacks a clear explanation of how it differs from other resistance strategies.
4. The introduction briefly mentions Puccinia triticina (Pt) isolates and salicylic acid (SA) as priming agents but does not provide a rationale for their selection or a review of their previous use in similar contexts.
5. Lack of Hypothesis Clarity. We hypothesize that pre-treatment of wheat plants with avirulent Pt isolates or exogenous SA will enhance defense responses against RWA infestation, as evidenced by reduced leaf damage and increased antioxidant enzyme activities. This study aims to evaluate the efficacy of these priming agents in inducing resistance and their potential as part of integrated pest management strategies in wheat production.
6. Figures:
Enhance figure legends by providing a more detailed description of the data presented, including the experimental conditions, the variables being compared, and the implications of the results.
Ensure that all figures follow a consistent format in terms of scale, labeling, and color coding to facilitate direct comparison of results. If different scales are necessary, this should be clearly indicated and justified in the figure legend.
7. Tables:
While the tables present mean values, there is no indication of the variability or statistical significance of the differences between treatments.
8. Include a comparative analysis of the results obtained from Pt isolates and SA treatments with those from conventional control methods or other biocontrol agents. This will help to contextualize the effectiveness of the priming agents within the broader scope of RWA management strategies.
9. Provide a more detailed interpretation of the variance analysis results, discussing the biological significance of the observed differences and interactions. This will help readers understand the complexity of the defense responses in wheat cultivars to RWA infestation under different priming conditions.
10. Include line graphs or bar charts to illustrate the changes in antioxidant enzyme activities. These visual representations will help readers to quickly grasp the trends and patterns in the data, especially the dynamics of enzyme activities following RWA infestation and priming treatments.
11. The discussion does not sufficiently address the limitations of the study. Provide a discussion on unexpected findings, exploring potential reasons for the observed outcomes and how they might relate to the current understanding of plant defense mechanisms. This will add depth to the results and provide a more comprehensive understanding of the data.
12. The authors should explicitly discuss how their results support or contradict the initial hypotheses. This will provide a clear narrative thread from the introduction, through the results, to the discussion.
13. The authors should include a more detailed discussion on the possible mechanisms by which Puccinia triticina isolates and salicylic acid influence the antioxidant response in wheat plants. This could involve speculation on the signaling pathways involved and how these might intersect with known defense responses. Discuss the findings in relation to other known priming agents to establish the relative efficacy of the tested compounds.
14. The authors should conclude the discussion by outlining future research directions that could address the gaps identified in their study; and discuss their findings in relation to other priming agents that have been used in similar contexts. This comparison will help to establish the relative efficacy and potential advantages of using Pt isolates and SA as priming agents.

Author Response
|
Reviewer Comments |
Author Comments |
|
Provide a full explanation of ROS (Reactive Oxygen Species) before its first mention. Clearly describe its role in biological systems, distinguishing between its signaling functions and potential oxidative damage |
“Reactive oxygen species” mentioned and its role highlighted |
|
The methodology for measuring SOD ( superoxide dismutase) activity lacks sufficient citation and explanation. Include a brief but clear description of the protocol used, referencing established methodologies |
The SOD activity mentioned sufficiently and cited. Citation highlighted with red color. |
|
Clarify why the protein factor is included in the formula for calculating SOD activity. Justify its necessity and relevance in the analysis |
To measure specific activity of separate antioxidants, I need to divide by Protein concentration. |
|
Provide a detailed description of the enzyme extract preparation. Include the source, buffer composition, and extraction procedure |
Section 2.2 describes extraction for SOD POD and CAT hence they have one protocol of extraction. Section 2.3 describes extraction protocol for APX.
|
|
Clarify why the protein factor is included in the formula for calculating SOD activity. Justify its necessity and relevance in the analysis |
Highlighted with red fount and cited [25] |
|
Correct the term protein, ensuring that terminology matches the context and methodology described |
Did |
|
Define hpi (hours post-infection) the first time it appears. This will ensure readers unfamiliar with the abbreviation understand its meaning immediately |
did |
|
Ensure uniform formatting of subsections and adhere to the journal’s structure guidelines for the methodology, results, and discussion sections. Consider rephrasing ambiguous sentences and simplifying overly complex phrasing to enhance readability. Verify all numerical data, figures, and tables for consistency and accuracy |
Tried to improve it
Thanks. |

Round 2
Reviewer 2 Report
Comments and Suggestions for Authors
Adjust the subtitles in the results section to align with the corresponding sections in the methodology. For example, 3.1. Seedling stage should be revised to (2.1.) Phenotyping of wheat plants prepared and not prepared for RWA infestation to ensure consistency and clarity.
Author Response
Adjust the subtitles in the results section to align with the corresponding sections in the methodology. For example, 3.1. Seedling stage should be revised to (2.1.) Phenotyping of wheat plants prepared and not prepared for RWA infestation to ensure consistency and clarity.
Response: I have adjusted and divided the seedling stage and the booting stage is sub-handing under "Phenotyping of wheat plants prepared and not prepared for RWA infestation" (3.1., 3.1.1., 3.1.2.).